# The ERG1A K^+^ Channel Is More Abundant in *Rectus abdominis* Muscle from Cancer Patients Than that from Healthy Humans

**DOI:** 10.3390/diagnostics11101879

**Published:** 2021-10-12

**Authors:** Sandra Zampieri, Marco Sandri, Joseph L. Cheatwood, Rajesh P. Balaraman, Luke B. Anderson, Brittan A. Cobb, Chase D. Latour, Gregory H. Hockerman, Helmut Kern, Roberta Sartori, Barbara Ravara, Stefano Merigliano, Gianfranco Da Dalt, Judith K. Davie, Punit Kohli, Amber L. Pond

**Affiliations:** 1Department of Surgery, Oncology and Gastroenterology, University of Padova, 35122 Padova, Italy; sanzamp@unipd.it (S.Z.); barbara.ravara@unipd.it (B.R.); stefano.merigliano@unipd.it (S.M.); gianfdada@gmail.com (G.D.D.); 2Department of Biomedical Sciences, University of Padova, 35122 Padova, Italy; marco.sandri@unipd.it (M.S.); roberta.sartori@unipd.it (R.S.); 3Anatomy Department, Southern Illinois University School of Medicine, Carbondale, IL 62902, USA; jcheatwood@siumed.edu (J.L.C.); lukebriananderson@gmail.com (L.B.A.); brittan.a.cobb@gmail.com (B.A.C.); 4Department of Chemistry and Biochemistry, Southern Illinois University School of Medicine, Carbondale, IL 62901, USA; rpb016@shsu.edu (R.P.B.); pkohli@chem.siu.edu (P.K.); 5Department of Epidemiology, Gillings School of Global Public Health, University of North Carolina at Chapel Hill, Chapel Hill, NC 27599, USA; cdlatour@live.unc.edu; 6Medicinal Chemistry and Molecular Pharmacology Department, Purdue University School of Pharmacy, West Lafayette, IN 47906, USA; gregh@purdue.edu; 7Physiko-und Rheumatherapie GmbH, 3100 St. Poelten, Austria; helmut@kern-reha.at; 8Biochemistry Department, Southern Illinois University School of Medicine, Carbondale, IL 62902, USA; jdavie@siumed.edu

**Keywords:** *erg1*a, potassium channel, dystrophin, sarcolemma membrane, cachexia, rhabdomyosarcoma

## Abstract

Background: The potassium channel encoded by the *ether-a-gogo-related gene 1A* (*erg1*a) has been detected in the atrophying skeletal muscle of mice experiencing either muscle disuse or cancer cachexia and further evidenced to contribute to muscle deterioration by enhancing ubiquitin proteolysis; however, to our knowledge, ERG1A has not been reported in human skeletal muscle. Methods and Results: Here, using immunohistochemistry, we detect ERG1A immunofluorescence in human *Rectus abdominis* skeletal muscle sarcolemma. Further, using single point brightness data, we report the detection of ERG1A immunofluorescence at low levels in the *Rectus abdominis* muscle sarcolemma of young adult humans and show that it trends toward greater levels (10.6%) in healthy aged adults. Interestingly, we detect ERG1A immunofluorescence at a statistically greater level (53.6%; *p* < 0.05) in the skeletal muscle of older cancer patients than in age-matched healthy adults. Importantly, using immunoblot, we reveal that lower mass ERG1A protein is 61.5% (*p* < 0.05) more abundant in the skeletal muscle of cachectic older adults than in healthy age-matched controls. Additionally, we report that the ERG1A protein is detected in a cultured human rhabdomyosarcoma line that may be a good in vitro model for the study of ERG1A in muscle. Conclusions: The data demonstrate that ERG1A is detected more abundantly in the atrophied skeletal muscle of cancer patients, suggesting it may be related to muscle loss in humans as it has been shown to be in mice experiencing muscle atrophy as a result of malignant tumors.

## 1. Introduction

Skeletal muscle atrophy is defined as a decrease in muscle contractile proteins that implies a weakening of strength. Atrophy occurs with normal aging and can also result from muscle disuse as well as muscle and/or neural damage or disease [1,2,3,4]. Age-related skeletal muscle atrophy (i.e., sarcopenia) has significant negative impacts on the health and quality of life for older persons [5]. The wasting syndrome (i.e., cachexia) that often affects cancer patients and other critically ill people (e.g., AIDS, sepsis, etc.) includes severe skeletal muscle atrophy and contributes to human morbidity and mortality [6,7,8,9]. Exercise is purported to be the most beneficial treatment for muscle atrophy [10,11,12]; however, many injured or ill persons are unable to participate adequately in this therapy. Additional strategies being explored to combat atrophy include nutritional therapy [2,10] and administration of pharmaceuticals such as proteolysis inhibitors [13], growth factors [14], beta-agonists [3], protein synthesis stimulators [15], and myostatin inhibitors [16]. Each has minimal efficacy and/or drawbacks, necessitating the development of more effective treatments. Therefore, an improved understanding of the mechanism(s) modulating atrophy is essential.

The *ether-a-go-go-related gene* (*erg1*) encodes a voltage-gated potassium channel known to contribute to the repolarization of the cardiac action potential in numerous mammalian species [17]. Alternative splice variants of *erg1 (erg1a* and *1b)* have been identified in both mice (mouse *erg1*, i.e., *Merg1a* and *Merg1b* [18]) and humans (human *erg1*, i.e., *Herg1a* and *Herg1b* [19]). HERG1A and MERG1A proteins are highly homologous full-length isoforms with 1159- and 1162-amino acids, respectively; MERG1A is 96% identical to HERG1A [18]. *Herg1b* and its mouse homolog *Merg1b* both have an alternate N-terminus sequence which results in ERG1b proteins which lack the first 342 amino acids of the ERG1a variant, having instead a different 36 amino acid sequence; ERG1b is otherwise basically identical to ERG1a in terms of remaining primary structure [18,19]. Expression of *erg1* and the ERG1 protein have been reported in numerous tissues of various species [20,21]; however, ERG1 detection is most prevalently reported and researched in the mammalian heart, where the ERG1 channel has been shown to be a hetero-multimer of both ERG1a and 1b [17,18,19,21,22,23].

Previously, we reported detection of the MERG1a isoform protein and *Merg1a* mRNA in skeletal muscle atrophied as a result of disuse (unweighting) and cancer cachexia, with detection of *merg1a* mRNA and MERG protein abundances being negligible in healthy muscle [24], as reported earlier by others [18]. Further, our studies have shown that expression of wild type (WT) *Merg1a* in the *Gastrocnemius* muscle of weight-bearing mice induces atrophy and that ectopic expression of a dominant-negative *Merg1a* mutant (*DN-Merg1a* [25]) suppresses atrophy in unweighted mice. Importantly, we have shown that ectopic expression of *Merg1a* in healthy mouse skeletal muscle increases the abundance of the skeletal muscle-specific ubiquitin–proteasome pathway (UPP) E3 ligase, MuRF1, and up-regulates UPP activity [24,26,27]. Here, using immunohistochemistry (IHC), we show, (to our knowledge) for the first time, that the ERG1 channel is detected in human *Rectus abdominis* (*RA*) muscle and that it is significantly more abundant in the *RA* of human cancer patients than in *RA* from both age-matched healthy control and healthy young adults. Additionally, using immunoblot, we show that ERG1 is more abundant in leg muscle samples taken (during above-knee amputations) from cachectic people than in those with a normal body mass index. Further, similar to the heart, we demonstrate in skeletal muscle that ERG1 immunofluorescence is localized in the sarcolemma.

## 2. Materials and Methods

Cell Culture. Mouse skeletal muscle-derived C_2_C_12_ cells and human rhabdomyosarcoma (RH30) cells were grown in Dulbecco’s Modification of Eagle’s Medium (DMEM; Thermo-Fisher, Maltham, MA, USA) supplemented with 10% heat-inactivated fetal bovine serum (FBS) and maintained in a humidified incubator with 10% CO_2_ at 37 °C. To differentiate C_2_C_12_ myoblasts into myotubes, myoblasts were grown in DMEM supplemented with 10% FBS until they reached ~85–90% confluence. The FBS medium was then replaced with DMEM medium supplemented with 2% heat-inactivated horse serum. Cells were incubated for 4 additional days to allow for terminal differentiation [28,29].

Human Skeletal Muscle. Muscle biopsies used for the immunohistochemistry were from healthy adult humans aged either 24–29 years (*n* = 4) or 61–86 years (*n* = 7) and from cachectic patients aged 65–88 years (*n* = 5; Table 1, Figure 1 and Figure 2). Samples were obtained during elective surgery by cold section of the *Rectus abdominis* muscle. Healthy subjects were control donors who were undergoing elective laparotomy for non-neoplastic and non-inflammatory diseases. Cachectic patients affected with colorectal, esophago-gastric or pancreatic cancers were enrolled for a project approved by the Ethical Committee for Clinical Experimentation of Padova (protocol number 3674/AO/15, 21 April 2016). All enrolled subjects were volunteers who signed written informed consent forms. Cachexia was diagnosed according to criteria described by Fearon et al. (2011) [30]. All biopsies were immediately frozen in pre-chilled isopentane and stored in liquid nitrogen until use. All enrolled subjects were volunteers who signed written informed consent forms. Muscle biopsies were de-identified before use. The skeletal muscle samples from limbs amputated just above the knees of patients with either a normal (>24) or low (<18) body mass index (BMI; *n* = 6; Table 2, Figure 3A–C) were purchased from The Collaborative Human Tissue Network (CHTN), Midwestern Division. All purchased samples were de-identified before approved use, and approval to use was waived by the SIU IRB.

Antibodies. For immunohistochemistry, the ERG1 antibodies (P9497 and AB5908) were purchased from Sigma (St. Louis, MO, USA) and diluted (see IHC methods below) prior to use. Similar results were obtained with each antibody, although the P9497 appeared to have a higher affinity for the ERG1 protein. The dystrophin (MAB1645MI) antibody was purchased from Thermo-Fisher Scientific (Waltham, MA, USA) and diluted 1:20 prior to use. The fluorescent secondary antibodies (goat anti-rabbit IgG antibody Alexa Fluor 488 and goat anti-mouse IgG antibody Alexa Fluor 568) were also purchased from ThermoFisher. For the immunoblot of human muscle, we used an “in-house” ERG1 antibody [31] and stained the membrane with Coomassie Blue once we had imaged the chemiluminescence (Sigma; St. Louis, MO, USA). The alkaline phosphatase-conjugated goat anti-rabbit IgG secondary antibody was purchased from Sigma.

Tissue Sectioning and Staining. Muscle sections were embedded using OCT™ (Electron Microscopy Sciences; Hatfield, PA, USA) and cryo-sectioned at 20 μm. Tissues were stained using either hematoxylin and eosin (H&E) to better identify cell structures or immunostained. H&E stain. Muscle sections were incubated in hematoxylin solution (Sigma GSH- 232, Merck Life Science S.r.l.) for 15 min and then rinsed with running tap water until the excess stain was removed. The sections were then dipped quickly in 0.5% Eosin B (Sigma E- 8761, Merck Life Science S.r.l.) for 1 min and immediately rinsed with distilled water. Slides were then dehydrated in ethanol solutions at increasing concentrations (70, 95, 100%) for 20 to 40 s and then in xylene for 1 min. Sections were finally covered using a cover-slip and Canada Balsam (Sigma C 1795, Merck Life Science S.r.l.) as a mounting medium. 

Tissue Immunohistochemistry. Isolated muscle fibers were fixed in 4%paraformaldehyde (as described above). Tissue sections were fixed in methanol at −20 °C for 10 min. Samples were then rinsed with PBS at room temperature (RT) and incubated in 0.3% H_2_O_2_ for 5 min, and this was followed by a rinse in RT PBS. All slides were then dipped for 5 s in filtered 0.3% Sudan Black (in 70% ethanol) to quench auto-fluorescence. These were then rinsed in PBS. At this point, the samples were incubated with blocking reagent I (10% normal goat serum (NGS; Sigma, St. Louis, MO, USA), 0.1% bovine serum albumin (BSA; Sigma) and 0.1% Tween-20 in PBS) for one hour at RT. These samples were then incubated overnight at 4 °C in either primary antibody diluted (as described above) in blocking reagent II (PBS with 5% NGS, 0.05% BSA, 0.1% Tween-20 and 0.1% sodium azide) or in blocking reagent II only as a control for primary antibody binding. The samples were rinsed in PBS containing 0.1% Tween-20 and incubated for 1 h at RT in a second primary antibody (where dual staining was desired) and then rinsed again in PBS containing 0.1% Tween-20. Samples were incubated in secondary antibody (as above) diluted 1:1000 with blocking reagent II. Finally, all slides were rinsed with PBS and mounted with Fluoromount G containing DAPI (Electron Microscopy Sciences; Hatfield, PA, USA). Immunostaining of Cultured Cells. C_2_C_12_ and RH30 cells were incubated on etched glass coverslips as described elsewhere [29]. The coverslips were rinsed in PBS, and the cells were fixed by incubation in cold methanol for 10 min. The cells were then rinsed, blocked as described above, and incubated in: (1) blocking buffer as control; or (2) ERG1 primary antibody (Sigma P9497) as described above. After incubation, the coverslips were rinsed with PBS and incubated for 1 h at room temperature in a secondary antibody: Alexafluor-568 conjugated goat anti-rabbit IgG (Thermo-Fisher; Waltham, MA, USA). Coverslips were then rinsed with PBS and mounted onto glass slides.

Imaging. Images of *Rectus abdominis* myofibers and of cultured cells were acquired using either a Leica DM4500 microscope with a Leica DFC 340FX camera or with a Zeiss Axioscop microscope connected to a Leica DC 300F camera. All sections compared within a study were immunostained together using the same reagents and timing. Acquisition parameters for all sections were maintained identically across all samples allowing reliable comparison of immunofluorescence intensities for different samples.

Quantitative Analysis and Statistics. Human Muscle Samples. For the staining in Figure 2 and Figure 3, treatment groups were: (1) young healthy adults, *n* = 4; (2) elder healthy adults, *n* = 7; and (3) cachectic elder adults, *n* = 5 (Table 1). For the H&E stains in Figure 2, random fields per one section per individual human were evaluated so that a total of >300 myofibers per subject were assessed. Human skeletal muscle biopsies are not easily orientable, especially when taken from pennate muscles. Consequently, muscle fibers are frequently kinked or distorted as a result of an oblique cut. To overcome this problem, we measured the lesser myofiber diameter instead of the cross-sectional area [32]. The muscle fiber diameter data were analyzed by a two-way ANOVA using the General Linear Model Procedure of SAS 9.4 (SAS Institute, Inc., Cary, NC, USA), and the means were compared using Tukey’s simultaneous confidence intervals. For Figure 3, images of two sections per individual human were evaluated by two independent observers who were blind to treatment groups. For each tissue section, three fields were imaged (thus, 6 fields per human), and the single point brightness was measured for 50 random consecutive points within the sarcolemma of each complete fiber within each field using ImageJ [33] with methods adapted from those published previously [34]. Brightness values were recorded as integers ranging from 0 (no signal) to 256 (white), and the average brightness value (±standard error of the mean, SEM) for each section was determined arithmetically. The resulting data were normalized to the young adult sample average and analyzed by two-way ANOVA using the General Linear Model Procedure of SAS 9.4 (SAS Institute Inc., Cary, NC, USA), with group and observer as factors in the model. First, we tested the significance of the interaction between group and observer. Because the interaction was non-significant, the term was removed, and the data from the two observers were combined per sample to represent a single replicate. All subsequent analyses were completed using a two-way ANOVA without the interaction term. Means were compared between groups using Tukey’s simultaneous confidence intervals to adjust for multiple comparisons. 

Tissue Culture. For Figure 4, Optical Density values of muscle sample proteins in the immunoblot were determined using ImageJ [33] and compared using a Student’s t-test. For all analyses, differences were considered significant for *p* < 0.05. For power analysis, we used the Power Procedure of SAS 9.4 (SAS Institute, Inc.). For Figure 5, brightness was determined as described earlier for Figure 3. Per each cell type (C2C12 and RH30), 6 etched cover glasses (*n* = 12) were treated with either primary antibody (3 cover glasses) or without primary antibody (3 cover glasses as control). Per cover glass, 10 cells were analyzed, and brightness data were analyzed by two-way ANOVA using the General Linear Model Procedure of SAS 9.4 (SAS Institute Inc., Cary, NC, USA). Means were compared between groups using Tukey’s simultaneous confidence intervals studies. 

Immunoblot. Membrane proteins were extracted from human skeletal muscles [35], and the protein content of each homogenate was measured using a DC Protein Assay Kit (Bio-Rad; Hercules, CA, USA). Aliquots of equal protein content were immunoblotted as described earlier [24]. Briefly, muscle homogenate samples (50 μg protein) were electrophoresed through a polyacrylamide gel (4–20% gradient), transferred to PVDF membrane (Bio-Rad; Hercules, CA, USA), and immunoblotted using an “in house” ERG1 antibody [31] and an Immun-Star Western Chemiluminescent Kit (Bio-Rad). Finally, the blotted membrane was stained with Coomassie Blue to confirm that membrane samples contained equal protein.

## 3. Results

ERG1 protein localizes to the sarcolemma membrane of human skeletal muscle. To explore ERG1 localization, human *Rectus abdominis* (*RA*) serial sections were immuno-stained (Figure 1). The absence of green or red fluorescence in the no primary antibody control treatment sections (Figure 1A) demonstrated that any red or green fluorescence in the other images is a product of primary antibody binding. Serial RA sections were also concurrently co-immunostained for ERG1 and the sarcolemmal membrane marker dystrophin. We detected dystrophin fluorescence (red) in the sarcolemmal membrane as expected (Figure 1B). Fluorescent confocal imaging also detected ERG1 fluorescence (green) in the sarcolemmal membrane of human RA muscle (Figure 1C). The appearance of a mild mottled green fluorescence pattern within the cytoplasm is consistent with localization also occurring within the cell. Merged images of the sections demonstrate that dystrophin and ERG1 both localize to the sarcolemmal membrane of human RA, although the absence of consistent yellow color from the merged fluorescent images suggests that the proteins do not localize in the same pattern (Figure 1D). Indeed, the ERG1 immunostaining appears to be punctate as has been described in the heart (Figure 1D inset) [22,23,25]. Further, the localization of both of these proteins to the sarcolemma is supported by merged images from which ERG1 and dystrophin fluorescence intensity profile measurements are discovered to follow a very similar pattern (Figure 1E) through the demarcated (white rectangle, Figure 1D) region of the sarcolemma. The Pearson correlation statistic of the profile data (red compared to green fluorescence intensities) for this single representative image was 0.96, demonstrating that the two fluorescent signals were similarly localized within the sarcolemmal membrane. The patterns noted here were similar for all human samples evaluated (*n* = 3 samples from cachectic cancer patients). The blue structures are DAPI stained nuclei.

ERG1 channel fluorescence is greater in human *Rectus abdominis* (*RA*) muscle from cachectic cancer patients than in age-matched healthy tissue. Human RA muscle samples were cryo-sectioned and stained with hematoxylin and eosin (Figure 2A–C). The morphology of the muscle from the healthy adults (both younger and elder) is similar, having an expected normal structure and average myofiber diameter. Indeed, there was no significant difference in the size of the fibers from the healthy people (Figure 2D). However, fiber sections from the elder cancer patients displayed fiber polymorphism with a smaller average fiber diameter, 38.8 ± 3.2 μm, being 27.6% smaller than those of the healthy age-matched controls and 28.2% smaller (*p* < 0.05) than those of the healthy young persons. Serial sections were further evaluated by immunostain procedures probing for the ERG1 channel protein (Figure 3). The absence of fluorescence in control sections immunostained without primary antibody (Figure 3A) demonstrates that any green fluorescence in the other sections is a product of primary antibody binding. Sections of RA from young and elder healthy people and from elder cancer patients were immunostained for ERG1 (Figure 3B,D). Indeed, ERG1 fluorescence was detected in the sarcolemma of all samples. (Note also the mottled green appearance of the fiber interiors, suggesting that ERG1 is also located in the cell interior.) The single point brightness was determined within the sarcolemma as described in the Methods section. Only a low amount of immunofluorescence intensity, representing the ERG1 protein, was detected in the sarcolemma of sections from young healthy adults (Figure 3B,E). The ERG1 immunofluorescence intensity was mildly more obvious in the sarcolemma of sections from the elder healthy adult samples (Figure 3C,E) but was markedly greater in the samples from older cachectic adults (Figure 3D,E). Indeed, when single point brightness within the sarcolemma was measured and compared, we found that ERG1 immunofluorescence intensity was 53.3% (*p* < 0.01) greater in the samples from cachectic patients than in the samples from age-matched healthy people (Figure 3E). When the ERG1 fluorescent intensity was compared in healthy young and healthy aged people, we discovered that there was a greater amount (10.6%) of ERG1 signal in the muscle of the healthy aged people than in that of the young; however, this difference was not statistically significant (Figure 3E).

The ERG1 channel protein is detected at a greater abundance in skeletal muscle from cancer patients than in that from healthy humans. The ERG1 protein is not an abundant protein in the heart and is even less abundant in normal skeletal muscle [22,23,24,31,36]; thus, the amount of human tissue necessary for an immunoblot is difficult to obtain, and isolation of the ERG1 protein is further complicated by how labile it is to proteolysis [24,31]. However, we were able to acquire skeletal muscle from humans undergoing leg amputations: three non-cancer patients (BMIs > 24) and three cancer patients (BMIs < 18). We prepared and concentrated membrane homogenates from the samples and immunoblotted 50 μg protein from each for ERG1 (Figure 4A). We then stained the PVDF membrane with Coomassie Blue to confirm that equal amounts of protein were loaded into each lane (Figure 4B). As previously shown with an immunoblot of heart muscle lysates digested with *N*-glycosidase [24,31], two glycosylated isoforms of ERG1 protein were detected in all of the muscles. We scanned the blots and used ImageJ32 to evaluate the optical densities (OD) of the protein bands. The data show that the average OD of the upper ERG1 protein band is a significant 61.5% (*p* < 0.05) greater in the tissue from the cancer patients than in the tissue from the healthy people (Figure 4A,C). The OD of the lower ERG1 protein band is 27.2% greater in the abnormal tissue than in the normal tissue; however, the *p*-value (*p* < 0.1) does not indicate statistical significance for this difference; nonetheless, it is still a low *p*-value and could be considered to be “approaching significance” (Figure 4A,C). It is likely that the difference would have been greater and possibly significant had more samples been available. Taken together, these results support the data presented in Figure 3, which demonstrates that the ERG1 protein is more abundant in cancer patients than in healthy subjects.

The ERG1 channel protein is more abundant in cultured human rhabdomyosarcoma cells than in cultured normal mouse muscle cells. Mouse C2C12 myoblasts and human rhabdomyosarcoma cells were cultured and immuno-stained for ERG1 protein (Figure 5). The absence of fluorescence in control sections immuno-stained without primary antibody (Figure 5A) demonstrates that any red fluorescence in the other sections is a product of primary antibody binding. Immunostaining reveals that both the non-malignant mouse C2C12 myoblasts (Figure 5A,B) and the human rhabdomyosarcoma cells (Figure 5C,D) present with ERG1 signal. However, when the ERG1 fluorescent intensities were compared (Figure 5E), we discovered that there was a statistically significant 52.2% (*p* < 0.05) greater amount of ERG1 signal in the human rhabdomyosarcoma cells than in the mouse C2C12 myoblasts.

## 4. Discussion

The detection of the ERG1 splice variant proteins, A and B, has been reported in the heart of numerous species, where they have been shown to form hetero-multimeric potassium channels that contribute to late-phase repolarization of the action potential [17,18,19,22,23]. ERG1A has also been detected in the skeletal muscle of rats and mice, with the highest abundance being in atrophic relative to normal muscle [24]. To our knowledge, this is the first report of the ERG1 protein in human skeletal muscle. Here, we demonstrate that the ERG1 protein is detected in the sarcolemmal membrane of the human *Rectus abdominis* muscle, showing a localization pattern similar to that of the sarcolemmal membrane marker dystrophin. The mottled appearance of fluorescence in the cell interior suggests that ERG1 protein is also located in the sarcoplasm of human skeletal muscle. This is not surprising because the ERG1 protein has been detected in conjunction with T-tubules in cardiac tissues [23,36]. Specifically, using immunohistochemistry and confocal microscopy, Rasmussen and colleagues (2004) [36] reported the detection of prominent ERG1 protein in the t-tubules of rat atria and ventricles, which correlated well with their detection of the known cardiac t-tubule marker protein dihydropyridine receptor (DHPR) α2-subunit; however, they did not report that the ERG1 and DHPR α2-subunit proteins co-localize within the structure. As expected, they also detected the DHPR α2-subunit protein in the peripheral sarcolemma along with the ERG1 protein, reporting relatively less prominent levels of the ERG1 protein in the sarcolemma (being concentrated near invaginations to t-tubules) than in the t-tubules. TEM confirmed these findings, revealing lower abundances of ERG1 protein in discretely localized clusters within the sarcolemmal membrane. Indeed, the presence of ERG1 clusters in the sarcolemma may explain the inconsistent localization pattern of ERG1 we find within the sarcolemma (Figure 1).

We then explored the relative abundances of ERG1 in the sarcolemmal membranes of *Rectus abdominis* muscle samples from young healthy persons, elder healthy patients, and elder cancer patients suffering from cachexia. Using images from immunostained tissues sections and single-point brightness data [33,34], we report that ERG1 fluorescence is statistically more abundant in the sarcolemma of the *Rectus abdominis* muscle of cachectic cancer patients than in the same muscle of healthy age-matched people. To further explore the abundance of ERG1 protein in cancerous tissue, we performed an immunoblot to compare ERG1 protein abundance in the skeletal muscle of healthy people versus cancer patients. This experiment was complicated by the fact that ERG1 protein is not a prominent protein in (even cardiac) muscle. It was further impacted by the fact that ERG1 is very labile to proteolysis [31,37,38,39]. Thus, it was difficult to get the necessary amount of human skeletal muscle to perform these studies, especially from young healthy people, and initially, we were unable to get enough human tissue from a consistent muscle or age group. However, we discovered that we could purchase skeletal muscle samples taken from people undergoing an above-the-knee amputation and that some of these were cancer patients (with low BMIs; Table 2, see Materials and Methods). We homogenized the human muscle samples and then concentrated the membrane fraction by adding a low volume of solubilization buffer to the final pellet. Indeed, using immunoblot, we found that the higher-mass ERG1 protein is statistically more abundant in the muscle of cancer patients than in that of healthy people. This finding concurs with our earlier immunoblot data showing that the mouse ERG1 proteins are significantly more abundant in the skeletal muscle of cachectic nude mice expressing malignant tumors than in the muscle of healthy control animals [24]. However, the *p*-value for comparison of the optical densities of the lower mass ERG1 protein is nearly equal to 0.1, being above the *p* < 0.05 level of significance set earlier; thus, the difference in abundance between the healthy and cancer patients for this ERG1 protein is not significant. Indeed, when we combine the OD data for both proteins and perform the ANOVA, we find that the difference is not statistically different (*p* < 0.08); nonetheless, it is low. Thus, because the upper band is statistically more abundant in the cancer patients than in the healthy ones and the difference in the lower band ODs is close to significance, we conclude that the increased ERG1 protein abundance in the muscle of cancer patients is important. One problem is that we have a very small sample size (*n* = 6). Therefore, we conducted a power analysis with our current combined ERG1 OD data to project what sample size would be necessary to detect a significant difference in ERG1 protein abundance between the elder healthy and elder cancer patients and found that an increase in sample size from a total of *n* = 6 to *n* = 12 would allow us to detect a difference at significance level *p* = 0.05 between the two groups with 90% power.

In earlier studies, we showed that mouse ERG1A protein was more abundant in the atrophic skeletal muscle of unweighted mice and demonstrated that the ERG1A K+ channel abundance increased in response to these atrophic insults and contributed to the proteolytic component of the skeletal muscle atrophy that followed [24]. Specifically, we over-expressed plasmid encoding mouse *Erg1a* in mouse Gastrocnemius muscles and demonstrated that (relative to controls) expression of the muscle-specific atrophy-related E3 ligase, *Murf1*, and overall UPP activity increased in response and that muscle fiber cross-sectional area significantly decreased in the *Merg1a*-expressing myofibers. We also inhibited atrophy in unweighted mice by (1) expressing a dominant-negative *Merg1a* plasmid [25] in their Gastrocnemius muscle and separately (2) treating the mice with the pharmacologic ERG1 blocker, astemizole (Sigma; St. Louis, MO). Because we find that ERG1 protein is more abundant in skeletal muscle from human cancer patients (Figure 3 and Figure 5), it is logical to suggest that ERG1 may also contribute to skeletal muscle atrophy in cachectic humans.

With our IHC single-point data, we also demonstrate that the ERG1 protein abundance is 10.6% greater in the *Rectus abdominis* muscle of healthy elder humans than in that of healthy young adults (Figure 3). Although the difference is not statistically significant, it does suggest that an increased abundance of ERG1 could be involved in the rather slow, natural muscle loss that occurs with age (i.e., sarcopenia [35]). One potential confounder is that our current analyses of immuno-stained muscle cross-sections evaluate fluorescence only from the sarcolemmal membrane because the immunofluorescence from t-tubules is not specifically identifiable in cross-sections. It is possible that if the analysis included the ERG1 protein in the t-tubules, the difference in ERG1 protein abundance detected in the tissues from the young people and those of the healthy older people could become more (or less) pronounced. Indeed, ERG1 protein was found to be more prominent in the t-tubules than in the sarcolemma of cardiac tissue [36]. Thus, we again conducted a power analysis with our current data to project what sample size would be necessary to detect a significant difference between the tissues from the young and aged healthy adults in terms of ERG1 fluorescence intensity and found that an increase in the sample size from a total of 11 (4 young and 7 healthy aged) to 46 would allow us to detect a difference at significance level *p* = 0.05 between the two groups with 90% power. Obviously, obtaining *Rectus abdominis* samples from 46 (especially 23 young, healthy) humans would be difficult; however, because the number is not completely unreasonable, it suggests that the difference in ERG1 protein abundance between the two groups may indeed be more important than the small sample size allows us to conclude. More interestingly, it leads us to suspect that the ERG1 protein would be even more abundant in the skeletal muscle of sarcopenic (i.e., aged and frail) people than in that of healthy aged individuals. Indeed, we showed that ERG1A protein is significantly more abundant in the Gastrocnemius muscles of aged (30 months) than in young (3 months) rats [40]. Another matter of note is that the ERG1 antibody we used does not allow us to distinguish between Erg1A and 1B alternative splice variants; however, we have not detected ERG1B protein or mRNA in mouse skeletal muscle [24], and in human skeletal muscle, we do not detect a protein consistent with ERG1B. Indeed, ERG1B is often considered to be “cardiac-specific” [18,19]; although there are reports of it being found in the brain [20] and some cancer cells [21]. To further test the potential importance of ERG1 in skeletal muscle, we immune-stained cultured human rhabdomyosarcoma cells and discovered that these cancer cells have a significantly greater ERG1 fluorescent abundance than C2C12 myoblasts, a non-malignant mouse muscle cell line. Indeed, ERG1 is detected in numerous cancer cell lines where it is suggested to modulate growth [21]. We conclude that the RH30 line would be a good in vitro model of ERG1 study in skeletal muscle.

An obvious question is: What does ERG1 do in skeletal muscle? Our earlier data suggest strongly that it modulates proteolytic activity in mouse skeletal muscle. Specifically, we showed that the over-expression of Merg1a in mouse skeletal muscle increases expression of the *Murf1* E3 ligase component of the UPP and enhances UPP activity [24], but interestingly, it does not affect the expression of the *Mafbx*/ATROGIN1 E3 ligase [26,27], which is considered to be an important marker of atrophy in skeletal muscle [41]. We do not yet know the full signaling mechanism for the enhanced *Murf1* expression or the increased MuRF1 protein levels that follow enhanced *Merg1a* expression [24,33,34]. Of course, in these studies, the *Merg1a* gene was over-expressed, and it can be argued that these results may not reflect true physiological effects. However, when the channel was blocked in unweighted mice, by both the expression of a dominant-negative subunit and the treatment with the known pharmacological ERG1 blocker, astemizole, the atrophy (decrease in the cross-sectional area) was inhibited [24]. These data support the idea that ERG1 may contribute to atrophy.

A more evident question is: Does ERG1 affect excitation–contraction coupling? ERG1 is a voltage-gated potassium channel shown to be located within the plasma membrane and the t-tubules of the heart [23,24]. Indeed, we have recently shown that increased HERG1A expression in C2C12 myotubes results in an increased intracellular calcium concentration [28]. However, the action potential duration in skeletal muscle is much shorter than in cardiac muscle (2–5 ms vs. 200–400 ms), and ERG1 has a very slow rate of activation, which is not likely to be relevant to action potential duration in skeletal muscle. So it is not likely that ERG1 affects the skeletal muscle action potential. Further, the relatively positive activation threshold for ERG1 (~−30 mV) [42] precludes its regulation of resting membrane potential (~−80 mV) [43] in skeletal muscle. However, ERG1 could uncouple excitation-contraction of skeletal muscle (as is known to occur with aging [35]) perhaps by (1) the modulation of DHPR activity to uncouple it from the ryanodine receptor or (2) perhaps more likely by inducing the proteolysis of proteins involved in excitation–contraction coupling. The location of the ERG1 protein in the t-tubules certainly places it in the proximity of the calcium release mechanism [23,36], suggesting that ERG1 may interact with/modulate this complex. Indeed, uncoupling of DHPR and ryanodine receptor proteins [44], progressive disorganization and spatial rearrangement of the excitation–contraction coupling apparatus [45], and abnormalities in calcium release [46] are shown to occur in skeletal muscle with aging [35] and disease [2]. Further studies are needed to determine whether an increase in the ERG1 channel expression contributes or responds to these effects.

## 5. Conclusions

Here, we report detecting significantly more ERG1 immunofluorescence and a greater ERG1 protein abundance in the sarcolemma of skeletal muscle sections from cancer patients than in muscle sections from age-matched or younger healthy humans. This is consistent with our earlier findings that show that ERG1 protein is more abundant in the atrophying muscle of cachectic nude mice expressing malignant tumors than in non-cachectic control animals without tumors [24]. We have shown that ERG1a modulates myofiber size in atrophying mouse muscle [24] and expect that it functions similarly in human muscle, but this needs to be explored. We also demonstrate that, as in cardiac tissue, ERG1 protein localizes to the sarcolemma. These data suggest that ERG1 protein is likely a marker for abnormal muscle growth and atrophy and perhaps even for skeletal muscle malignancy. Because skeletal muscle health is a prominent health concern for cancer patients as well as injured, ill, and elderly people, it is important that research on the skeletal muscle ERG1 K+ channel continue.

## Figures and Tables

**Figure 1 diagnostics-11-01879-f001:**
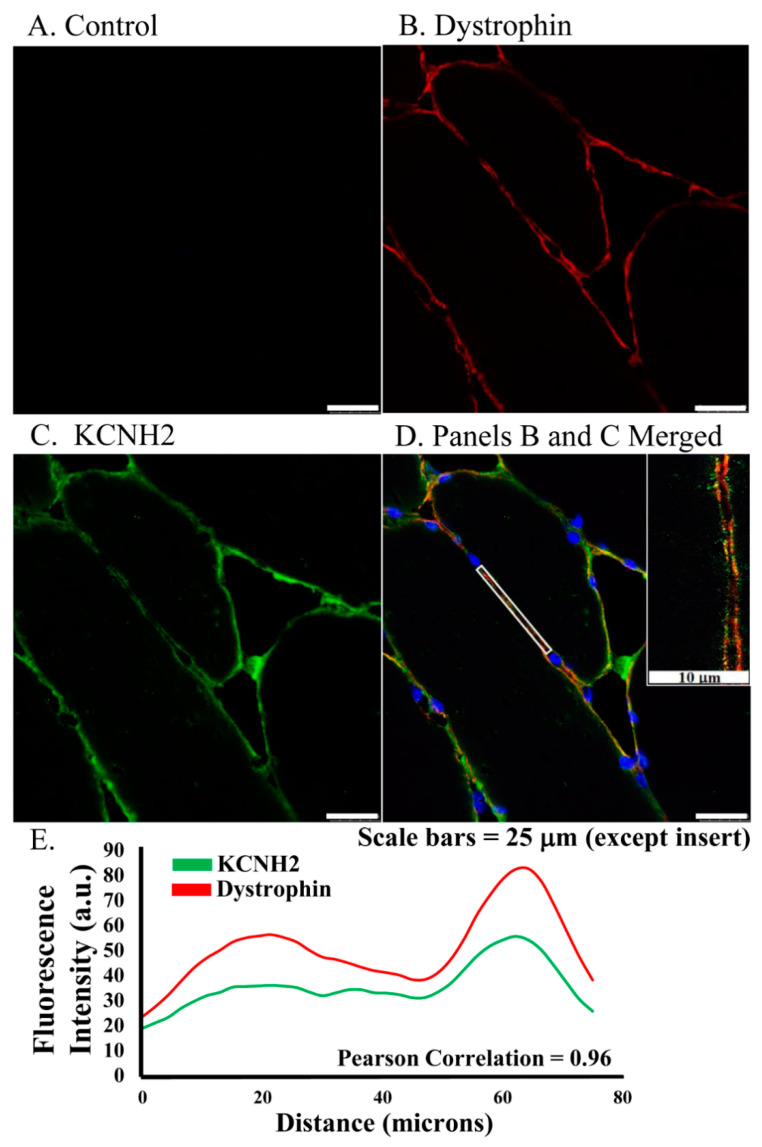
ERG1 fluorescence localizes to sarcolemma of human *Rectus abdominis* myofibers as shown by fluorescent immunohistochemistry of muscle from older healthy adults. (**A**) Fluorescence was absent from control sections immuno-stained without primary antibody. (**B**) Fluorescence (red) of dystrophin, a skeletal muscle sarcolemma marker, was detected in sarcolemma as expected. (**C**) ERG1 fluorescence (green) was detected in sarcolemma. (**D**) ERG1 (green) and dystrophin (red) immunofluorescence were both detected in the sarcolemma of human Rectus abdominis muscle. The white rectangular region is the area in which the fluorescence intensity (**E**) was measured. The inset image is taken from within the white rectangle. DAPI staining (blue) revealed nuclei. Scale bar is 25 μm except for inset where scale bar is 10 μm. Samples were stained and imaged using the same parameters in a single setting. (**E**) ERG1 (green) and dystrophin (red) fluorescences both localize to the sarcolemma as evidenced by fluorescence intensities, which follow the same pattern through the demarcated (white rectangle) region of the sarcolemma of this representative section. Images were acquired using a Leica DM4500 microscope with a Leica DFC 340FX camera.

**Figure 2 diagnostics-11-01879-f002:**
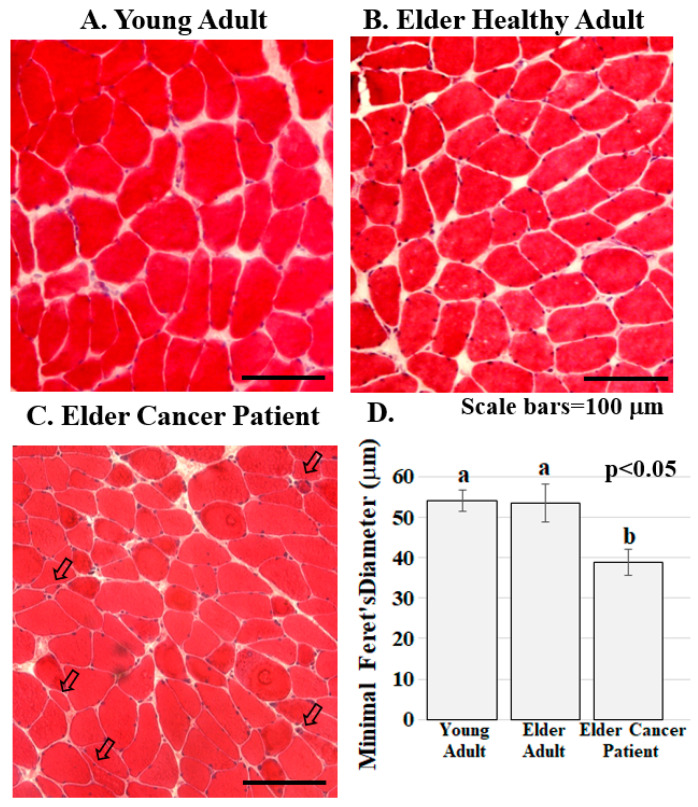
Skeletal muscle from cancer patients with low BMI present with abnormalities. (**A**–**C**). Human RA muscle sections stained with hematoxylin and eosin demonstrate that muscle structure and diameter are normal in the samples from the healthy young (**A**) and healthy elder (**B**) people while muscle from the elder cancer patients (**C**) had decreased myofiber diameter, severely atrophic, flat shaped or angulated myofibers (arrowed). Scale bars are 100 μm; (**D**) Muscle fiber diameter is significantly lower in elder cancer patients relative to healthy young and age-matched adults. Graph bars represent mean minimal fiber diameter (μm) ± the standard error of the mean. Different lower-case letters represent statistical difference (*p* < 0.05). All samples were stained and imaged using the same parameters in a single setting. Graph bars represent mean minimal Ferret’s diameter (μm) ± the standard error of the mean. Different lower-case letters represent statistical difference (*p* < 0.05). Images were acquired using a Zeiss Axioscop microscope connected to a Leica DC 300F camera.

**Figure 3 diagnostics-11-01879-f003:**
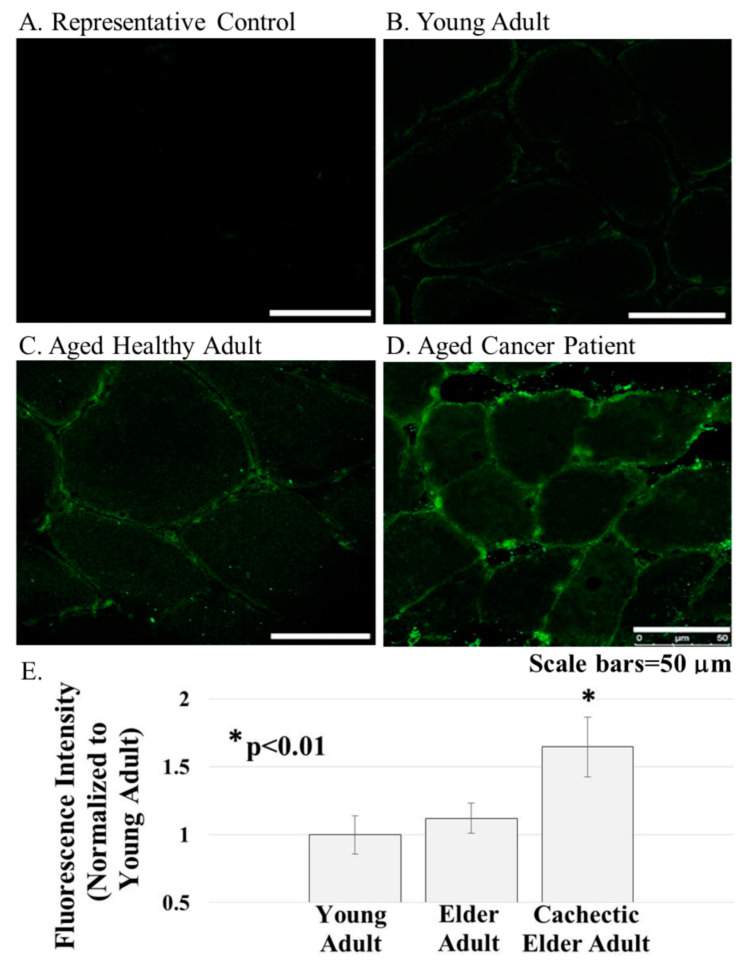
ERG1 protein was detected in cross-sections of the human *Rectus abdominis* muscle by fluorescent immunohistochemistry. (**A**) Fluorescence was absent from control sections immunostained without primary antibody; (**B**–**E**) The ERG1 fluorescence was greater in the skeletal muscle from elder cancer patients ((**D**), *n* = 5) than in that of healthy aged-matched ((**C**), *n* = 7) or young adult ((**B**), *n* = 4) humans. All samples were stained and imaged using the same parameters in a single setting. Scale bars are 50 μm. E. ERG1 fluorescence is statistically more abundant in the skeletal muscle of elder cancer patients. Graph bars represent mean fluorescence intensity ± the standard error of the mean. The “*” designates a statistical difference (*p* < 0.01). Images were acquired using a Leica DM4500 microscope with a Leica DFC 340FX camera.

**Figure 4 diagnostics-11-01879-f004:**
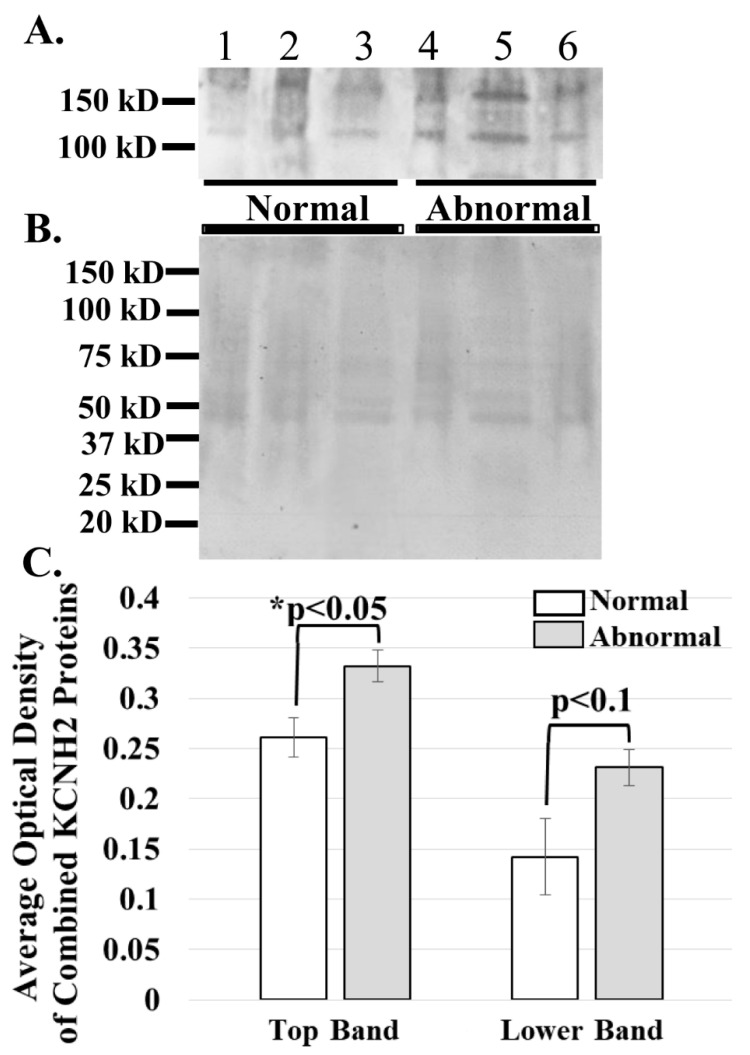
ERG1A protein variants are more abundant in the skeletal muscle of cancer patients than in that of healthy humans. (**A**) Human limb skeletal muscle membrane homogenates were prepared from both cancer patients (Abnormal) and healthy (Normal) humans and immunoblotted (50 μg) to show that the ERG1A protein is more abundant in the muscle from cancer patients than in healthy individuals; (**B**) Coomassie stain of the immunoblotted membrane reveals that equal amounts of membrane homogenate protein were loaded into the wells for SDS-PAGE; (**C**) The graph demonstrates the difference in mean optical densities (OD) of the ERG1A proteins. The bars represent the average OD ± the standard error of the mean of either the top or lower (as indicated) ERG1 protein bands in Panel A. The “*” represents a statistical difference (*p* < 0.05).

**Figure 5 diagnostics-11-01879-f005:**
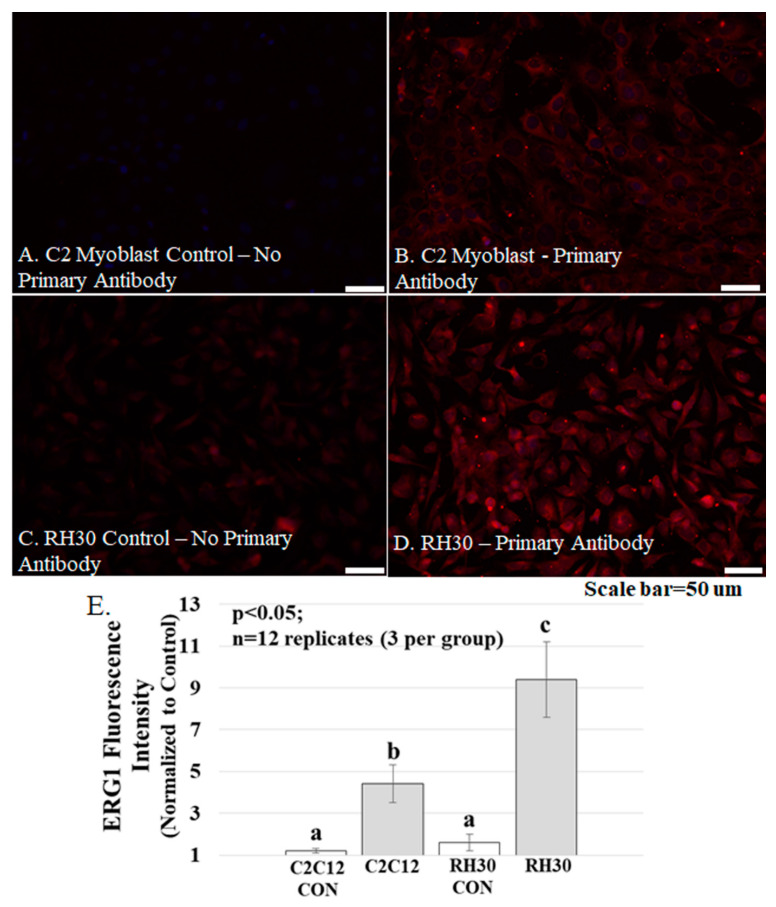
ERG1 protein fluorescence is more abundant in human rhabdomyosarcoma (RH-30) cells Table 2. C12 cells. (**A**–**D**) Fluorescence was absent from C2C12 (**A**) and RH30 (**C**) control cells immunostained without primary antibody but was evident in the C2C12 (**B**) and RH30 (**D**) cells stained with primary ERG1 antibody. Scale bars represent 50 μm. All samples were stained and imaged using the same parameters in a single setting; (**E**) ERG1 fluorescence is statistically more abundant in RH30 cells than in non-cancerous C2C12 cells. Graph bars represent mean fluorescence intensity ± the standard error of the mean. Different lower-case letters represent statistical difference (*p* < 0.05). Images were acquired using a Leica DM4500 microscope with a Leica DFC 340FX camera.

**Table 1 diagnostics-11-01879-t001:** Human subjects providing samples for immunohistochemistry.

Sample Source	Tumor Site	Sex	Age (yrs)	BMI
Cachectic patients; *n* = 5; Mean Age ± SD = 68.2 ± 11.4 yrs				
1	Esophagus	M	65	23.7
2	Colon	M	88	25
3	Pancreas	F	66	21.5
4	Pancreas	F	59	24.7
5	Pancreas	F	63	20
Healthy aged; *n* = 7; Mean Age ± SD = 67.9 ± 9.1 yrs				
6		M	63	30
7		M	62	24.7
8		M	68	24.6
9		M	61	22.7
10		F	62	nd
11		F	86	24.6
12		F	73	29.6
Healthy young; *n* = 4; Mean Age ± SD = 25.8 ± 2.2 yrs				
13		M	24	
14		F	29	
15		F	25	
16		F	25	

Abbreviations: yrs = years; *n* = number of subjects; nd = not determined; SD = standard deviation.

**Table 2 diagnostics-11-01879-t002:** Human subjects providing samples for immunoblots.

Sample Source	Tumor Site	Sex	Age (yrs)
Limb Tissue from Patients with BMI > 25; *n* = 6; Mean Age ± SD = 60 ± 3.6 yrs			
1	None	F	58
2	None	F	67
3	None	M	55
Limb Tissue from Cancer Patients with BMI < 18; *n* = 3; Mean Age ± SD = 63.7 ± 0.3			
4	Arm	F	63
5	Esophagus	M	64
6	Tumor Site Not Reported	M	64

Abbreviations: yrs = years; *n* = number of subjects; SD = standard deviation.

## Data Availability

The datasets analyzed for this study are available from the corresponding author upon request.

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
