# Peer review of "The ERG1A K+ Channel Is More Abundant in Rectus abdominis Muscle from Cancer Patients Than that from Healthy Humans"

_diagnostics, 2021, doi:10.3390/diagnostics11101879_

Round 1

Reviewer 1 Report

In this paper the authors claim that they for the first time detected ERG1 channel protein in a human skeletal muscle and  in particular in the Rectus abdominis muscle. To obtain their results they performed IHC and single points bright data experiments using biopsies from healthy young or elder patients as well as from cancer patients. The second novelty disclosed from the results of the paper is that ERG1 protein is more present in cancer cachetic patients than in healthy. Similarly  healthy elder patient tend to express ERG1 more than healty young patients.

The issue investigated in the paper is conceptually simple and the experiments are limited from the number of available biopsies, as the authors themselves state in relation to the statistics of the experiments (see line 330). In order to partially compensate for the above limitations the authors check ERG1 protein expression in a cell line of normal skeletal muscle and from a rabdomyosarcoma line, essentially confirming results from biopsies.

Overall the manuscript is clear and well written and the results seems mostly reliable. Thus in my opinion only minor revisions can be performed to render the manuscript acceptable:

  • Could the authors explain somewhere how do they choose the myofiber diameter to measure?
  • Line 93 : please correct FBS medium that has no been defined before        
  • Line 114 : please define BMI
  • Line 115: please correct  hemotoxylin
  • In figure 3C the dimension of myofiber don’t correspond sufficiently well with the value of the histogram. Could the author check and correct the image or the histogram?
  •  Line 447 : please correct “muscle24”
  •  Line 454 : please correct “growth21”
  •  Line 471: ERG1 is not “membrane bound” :please correct in “plasma membrane” for instance

Author Response

Dear Reviewer #1,

Thank you for the timely and thorough review of our manuscript.  We greatly appreciate your time and the effort you have put forth to help us improve our paper.  We have responded to your guidance as outlined below:

  1. Reviewer. “Could the authors explain somewhere how did they choose the myofiber diameter to measure?”

Response: We have updated our Materials and Methods section to include the following statement in the section labeled “Quantitative Analysis and Statistics.  Human Muscle Samples.” (highlighted in yellow on page 4):

“Human skeletal muscle biopsies are not easily orientable, especially when taken from pennate muscles.  Consequently, muscle fibers are frequently kinked or distorted as a result of an oblique cut.  To overcome this problem we measured the lesser myofiber diameter instead of cross-sectional area [32].”

This required the addition of a reference (number 32) and we have revised the paper accordingly.

  1. FBS is defined as fetal bovine serum in the “Cell Culture” section of Materials and Methods on page 3.
  2. BMI is defined as body mass index in the “Human Skeletal Muscle” section of Materials and Methods on page 3.
  3. Hematoxylin is spelled either “hematoxylin” or “haematoxylin” in the United States; we have chosen to spell it in the simpler manner.
  4. Reviewer. “In figure 3C the dimensions of myofibers don’t correspond sufficiently well with the value of the histogram. Could the author check and correct the image or the histogram?”

Response:  The data reported in the histogram are correct. In the skeletal muscle of cachectic patients, we observed a huge heterogenicity in myofiber size as a result of high-grade myofiber polymorphism, with some having a normal size and round shaped morphology while a majority of them were severely atrophic, flat shaped or angulated. We included a new panel C in the Figure 3 in which these morphological features can be better appreciated.

  1. Both “muscle24” and “growth21” have been corrected and now read: “muscle [24]” and “growth [21].”

  1. Reviewer. “Line 471: ERG1 is not “membrane bound” :please correct in “plasma membrane” for instance.” 

Response:  An appropriate change has been made.

“A more evident question is: Does ERG1 affect excitation contraction coupling?  ERG1 is a voltage-gated potassium channel shown to be located within the plasma membrane and the t-tubules of heart [23,24].” 

Thank you again!

Amber Pond

Reviewer 2 Report

Mistypings:

  1. In all the Figure legends there are spiral symbols. I think this is a kind of misprinting and it should be a “micro symbol”. In the “Results” part I also found this spiral symbol.
  2. In the legend of Figure 2, the panel identifier for (D) is missing.
  3. In the “Results” part, in the first sentences the words start with capital letter. Did this happen by accident, or did you consciously write it like that?
  4. In line 338, I think you accidentally wrote “green” instead of “red”.
  5. In line 409, 447, 454 and 463, the square bracket is missing around the reference numbers.
  6. Accidentally, the term "6. Patents " was added to line 503.

Questions:

  1. If the intensity of fluorescence was normalized to the "young adult" value, how is it possible that the value of "young adult" is not 1 in Figure 3, panel E?
  2. In Figure 4 panel C, what do “top band” and “lower band” mean? Do these bands refer to the bands in panel A?
  3. Similarly to Figure 3, if the fluorescence intensity was normalized to the control value, then the control values should be 1 in Figure 5 panel E, right? Or maybe I misunderstood something…
  4. It is possible that the ERG1 ion channel does not have a canonical function in the cell, i.e. ERG1 does not act by conducting ion currents through the cell membrane, but by somehow exerting its effect in a different way. It might also be nice to perform electrophysiological measurements on these cells to see if the ERG1 channels are functional.

It would also be worthwhile to do some functional experiment to check if inhibition of the ERG1 channel with selective ERG1 inhibitors creates some kind of change in the functionality of these cells. But I'm afraid these experiments go beyond what the authors want to communicate.

Overall, my view is that the research design is appropriate, and the applied techniques are adequated. The results are usually clearly presented by quality figures.

Author Response

Dear Reviewer #2,

Thank you for the timely and thorough review of our manuscript.  We greatly appreciate your time and the effort you have put forth to help us improve our paper.  We have responded to your guidance as outlined below:

Mistypings: 

  1. Indeed, the spirals were supposed to be the “micro” symbol. These have been corrected.
  2. The caption describing panel D of figure 2 has been added to the legend and is highlighted in yellow on page 16.
  3. In the “Results” section, the words in the first sentences begin with capital letter by design. It is to designate that this line is a descriptor of the text that follows.  We can change it if it is deemed inappropriate. 
  4. Indeed, the word “green” should be “red.” This has been corrected and is highlighted in yellow on page 7.
  5. Brackets have been added to encompass these reference numbers.
  6. The “6. Patents” statement has been removed from line 503.

Questions:

  1. If the intensity of fluorescence was normalized to the "young adult" value, how is it possible that the value of "young adult" is not 1 in Figure 3, panel E?

Response:  Indeed the value of the “young healthy” fluorescence intensity should be exactly 1.0.  The graph has been corrected to demonstrate this.

  1. In Figure 4 panel C, what do “top band” and “lower band” mean? Do these bands refer to the bands in panel A?

Response:  Yes, the top and lower bands refer to the ERG1 protein bands in Panel A.  The figure legend has been revised to better describe this and the changes have been highlighted in yellow on page 16.

The bars represent the average OD ± the standard error of the mean of either the top or lower (as indicated) ERG1 protein bands in Panel A. 

  1. Similarly to Figure 3, if the fluorescence intensity was normalized to the control value, then the control values should be 1 in Figure 5 panel E, right? Or maybe I misunderstood something…

Response:  These are raw intensity data and have not been normalized – although the Y-axis on the figure was labeled “normalized.”  The axis label has been corrected and the Materials and Methods section is correct, not describing any normalization procedures.

  1. It is possible that the ERG1 ion channel does not have a canonical function in the cell, i.e. ERG1 does not act by conducting ion currents through the cell membrane, but by somehow exerting its effect in a different way. It might also be nice to perform electrophysiological measurements on these cells to see if the ERG1 channels are functional.

Response:  YES!  It is very possible that the channel affects cell function by modulating messenger signaling, specifically through one or some of the signaling domains known to exist on its N-terminus.  Indeed, we have shown that ERG1 increases calcium levels in C2 cells*, but have not yet published work fully describing the mechanism.  EP experiments to show presence or absence of ERG1 current conduction in the rhabdomyosarcoma cells are planned along with studies to determine the effect of current block.  This is beyond the scope of this current paper.

*Whitmore C, Pratt EPS, Anderson LB, Bradley K, Latour SM, Hashmi MN, Urazaev AK, Weilbaecher R, Davie JK, Wang W-H, Hockerman GH, Pond AL.  The ERG1a potassium channel increses basal intracellular concentration and calpain activity in skeletal muscle cells.  Skeletal Muscle. 2020;10(1).  DOI: 10.1186/s13395-019-0220-3.

  1. It would also be worthwhile to do some functional experiment to check if inhibition of the ERG1 channel with selective ERG1 inhibitors creates some kind of change in the functionality of these cells. But I'm afraid these experiments go beyond what the authors want to communicate.

Indeed, we plan electrophysiological experiments and work with ERG1 blocking compounds as well as studies of the effect of ERG1 on membrane properties using atomic force microscopy.  We believe that this work is beyond the scope of this current paper.

Thank you!

Amber Pond
